# An Image Compression Encryption Algorithm Based on Chaos and ZUC Stream Cipher

**DOI:** 10.3390/e24050742

**Published:** 2022-05-23

**Authors:** Xiaomeng Song, Mengna Shi, Yanqi Zhou, Erfu Wang

**Affiliations:** Electrical Engineering College, Heilongjiang University, Harbin 150080, China; 2191283@s.hlju.edu.cn (X.S.); 2191310@s.hlju.edu.cn (M.S.); 2191294@s.hlju.edu.cn (Y.Z.)

**Keywords:** ZUC stream cipher, compressed sensing, chaos, image encryption

## Abstract

In order to improve the transmission efficiency and security of image encryption, we combined a ZUC stream cipher and chaotic compressed sensing to perform image encryption. The parallel compressed sensing method is adopted to ensure the encryption and decryption efficiency. The ZUC stream cipher is used to sample the one-dimensional chaotic map to reduce the correlation between elements and improve the randomness of the chaotic sequence. The compressed sensing measurement matrix is constructed by using the sampled chaotic sequence to improve the image restoration effect. In order to reduce the block effect after the parallel compressed sensing operation, we also propose a method of a random block of images. Simulation analysis shows that the algorithm demonstrated better encryption and compression performance.

## 1. Introduction

Under the background of the rapid development of data network and information technology, the application of image is increasingly extensive. Image encryption is one of the common means to protect image information [1]. The classical cryptographic encryption algorithms, including DES(Data Encryption Standard) and AES(Advanced Encryption Standard),mainly deal with text information, and the processing efficiency is not high. Traditional encryption algorithms cannot achieve efficient encryption of images.

In recent years, many image encryption schemes have been proposed successively, such as chaos theory [2,3,4,5], the DNA method [6,7,8,9], cellular automata [10,11,12,13] and so on. The article published by Fridrich in 1997 used chaos theory for the first time [14]. He designed an image-encryption algorithm and constructed a classical scrambling-diffusion image encryption framework. Chaos can be divided into two categories: high-dimensional chaos and one-dimensional chaos [15]. A one-dimensional chaotic system is a simple function with a low computational cost.

Due to its simplicity and efficiency, these have been used in various encryption algorithms [16,17,18]. However, they have the disadvantages of a small key space, single parameter and weak randomness of the generated chaotic sequence [19,20]. The high-dimensional chaotic structure is complex and is more sensitive to the initial values and parameters. However, it is difficult to implement software and hardware due to its complex structure and huge iterative calculation amount [21]. Therefore, the algorithm in this paper mainly uses a one-dimensional chaotic system to ensure the efficiency of encryption.

At first, scholars made efforts in the design of image-encryption algorithms based on one-dimensional chaotic systems. For example, in 2012, Wang et al. proposed a chaotic image encryption scheme, in which only the classical Logistic system was used to scramble and diffuse images [22]. In the following year, Arroyo successfully cracked the scheme of Wang by choosing plaintext attack [23]. It is shown that this method of scrambling and diffusing images with a one-dimensional chaotic system has a small key space and low sensitivity of plaintext.

In recent years, people have begun to use treatments on one-dimensional chaotic systems to improve the chaotic performance. PakC et al. used the difference between the one-dimensional chaotic map output sequence and its rounded and amplified output sequence to construct a new system with better performance and a wider chaotic range [24]. In 2018, Lan et al. used three one-dimensional chaotic maps to construct two more chaotic integrated chaotic systems and performed cascade, nonlinear combination and switching operations [25].

In the same year, Parvaz et al. combined Logistic, Sine and Tent chaotic maps to construct a combined chaotic system to perform multiple rounds of cyclic shift encryption on images [26]. This method of combining multiple one-dimensional chaotic systems to form a new chaotic system not only improves the performance of the original chaotic system but also ensures the operation efficiency.

Image information faces the dual pressures of security and resource constraints during storage and transmission. The theory of compressed sensing (CS) [27] realizes the simultaneous encryption and compression of the image and reconstructs the signal through the convex optimization solution process. Many scholars have implemented image encryption using compressed sensing [28,29,30,31].

It should be indicated that, when using compressed sensing to encrypt an image, in essence, the measurement matrix is used as the encryption key, and the corresponding measurement result is used as the ciphertext. This is essentially a linear measurement process so that it cannot resist known-plaintext attacks and chosen-plaintext attacks. It is necessary to combine compressed sensing with traditional diffusion encryption algorithms to ensure sufficient security of the encryption scheme [32]. By further improving the sparse basis, measurement matrix and reconstruction algorithm in compressed sensing theory, the quality of the restored image under the same compression rate can be effectively improved.

This paper uses a one-dimensional chaotic map to generate a measurement matrix for compressed sensing. The ZUC cryptosystem [33] is introduced to improve the shortcomings of the one-dimensional chaotic system, which has a small key space and insufficient randomness.The algorithm adopts parallel compressed sensing technology as a whole. First, the sparse coefficient matrix of the image is obtained by discrete wavelet transform, and then it is randomly divided into blocks. Then, we set the initial chaotic value and use the one-dimensional Sine–Tent–Logistic (STL) chaotic map to iteratively generate the chaotic sequence.We use the ZUC stream cipher system to sample the STL sequence to obtain a new chaotic sequence.

In the compressed sensing stage, the new chaotic sequence is used to construct the compressed sensing measurement matrix and parallel compressed sensing processing is performed on the divided matrix. Finally, subsequent scrambling and diffusion operations are performed on the compressed matrix. The image is reconstructed using the Orthogonal Matching Pursuit (OMP) algorithm at the decryption stage.

The main contributions of this algorithm are:

(1) Randomly partition the sparse coefficient matrix of the image to reduce the pixel correlation of the image. (2) Use the chaotic initial value as the key for encryption and decryption to avoid the waste of resources caused by transmitting the entire measurement matrix. (3) After the image is divided into blocks and encrypted in parallel, the time required for image encryption can be reduced, and the encryption efficiency can be improved. (4) Using the ZUC stream cipher system to sample the one-dimensional chaotic sequence, the randomness of the one-dimensional chaotic system is improved, and the key space is enlarged.

The remainder of this article is as follows. Section 2 introduces the fundamental knowledge involved in the proposed encryption algorithm, mainly including compressed sensing, STL chaotic map and ZUC stream cipher. Section 3 introduces the sampling method of a one-dimensional chaotic system and the effect after sampling. A image random block method is also proposed in this section. Section 4 summarizes the specific details of the proposed encryption scheme. Section 5 presents the simulation results of the algorithm and analyzes its performance. Section 6 presents our conclusions and summarizes the full text.

## 2. Fundamental Knowledge and Related Technologies

This section mainly introduces the basic theories and techniques involved in the proposed algorithm, including the compressed sensing technology, STL chaotic map and ZUC stream cipher system. The main purpose of this section is to facilitate the description of the encryption schemes that follow.

### 2.1. Parallel Compressed Sensing

CS breaks through the limitation of the traditional Nyquist sampling theorem. Parallel compression sensing technology [34] is divided into compression measurement and reconstruction. Signal compression is achieved through linear measurement, and signal reconstruction is achieved through sparse prior characteristics of signals. Divide an image of size M×N into *K* sub-blocks $x_{i}$ of size M×n, K=N/n. Represented as follows:(1)$$x_{i}=\Psi s_{i},$$
where Ψ is a sparse basis matrix. The block operation can perform the process of image compression measurement, and reconstruction can be performed in parallel, which can reduce the image processing time. The image is measured in columns, and the measurement process can be expressed as:(2)$$y_{i}=\Phi x_{i}=\Phi \Psi s_{i},$$
where Φ is the measurement matrix of size m×M. m=round(r×M), and *r* is the compression ratio. The measured value of the whole image can be expressed as:(3)$$Y=[y_{1},y_{2},\cdots ,y_{n}],$$

In the process of image restoration, the sparse sparse matrix of the original image can be recovered by solving the L1 norm problem as long as it meets the requirements of Φ and Ψ is not relevant. In the process of image restoration, the complicated operation of inverting the matrix is avoided. This process can be described as follows:(4)$${\hat{s}}_{i}=\arg\min_{s_{i}\in R^{N}}\;∥s_{i}∥\;s.t.\;{\hat{y}}_{i}=\Phi \Psi s_{i},$$

Commonly used measurement matrices include the Gaussian random matrix, partial Adama matrix and measurement matrix constructed by a chaotic sequence, etc., [35]. There are two reconstruction methods of compressed sensing [36]. One is to seek a minimum $l_{0}$ norm based on a greedy iterative algorithm, including matching pursuit (MP), orthogonal matching pursuit (OMP), etc. The other is a convex optimization algorithm to find the $l_{1}$ norm, including the base tracking method, gradient projection method, etc. In the proposed scheme, the OMP algorithm is used to reconstruct the image.

### 2.2. STL Chaotic Map

Zhou proposed a method of a cascading Tent and Logistic chaotic system to improve the performance of a one-dimensional chaotic system [37]. On this basis, Pounma et al. proposed STL (Sine–Tent–Logistic) chaotic system. This is a one-dimensional system combining the above three chaotic systems [38]. The Sine map, Tent map and Logistic map are shown in (Equation 5).
(5)$$\left\{\begin{array}{c} S_{n+1}=\mu_{S}\sin\left(\pi S_{n}\right);\;0\leq \mu_{S}\leq 1,\;S_{n}\in \left[0,1\right]\\ T_{n+1}=1-\mu_{T}\left|T_{n}-0.5\right|;\;0\leq \mu_{T}\leq 2,\;T_{n}\in \left[0,1\right]\\ L_{n+1}=\mu_{L}L_{n}\left(1-L_{n}\right);\;0\leq \mu_{L}\leq 4,\;L_{n}\in \left[0,1\right] \end{array}\right.,$$
where $\mu_{S}$, $\mu_{T}$ and $\mu_{L}$ are chaotic parameters. The steps of constructing the STL system are as follows:

(1) Fusion: Since Tent mapping shows highly chaotic behavior when the parameters are controlled, the control parameter is fixed, and the Logistic and Tent seed mappings are fused to generate a new mapping;

(2) Cascading: Use the Sine mapping and the mapping generated in Step (1) to perform cascading. As the Sine mapping shows highly chaotic behavior, the control parameter of the Sine mapping in the cascading step is set to 1.

Figure 1 shows the STL map structure. The fusion operation combines Tent and Logistic seed maps in a nonlinear way. $C_{n}$ was input to the two seed maps, and we add their outputs. The cascade calculation connects Sine mapping with Tent and Logistic mapping and finally obtains $C_{n+1}$.

The specific mathematical representation of the STL system [38] is shown in (6).
(6)$$C_{n+1}=\left\{\begin{array}{c} \sin\left(\pi \left(\left(1-2\left|C_{n}-0.5\right|\right)+\left(\mu C_{n}\left(1-C_{n}\right)\right)\right)\right)-1,C_{n}>1\\ \sin\left(\pi \left(\left(1-2\left|C_{n}-0.5\right|\right)+\left(\mu C_{n}\left(1-C_{n}\right)\right)\right)\right)+1,C_{n}<0 \end{array}\right.\;\mu >0,$$

Figure 2 for the bifurcation diagram of the STL chaos mapping shows that the STL map produced by the chaos and chaotic sequence is highly uniform in distribution.

The traditional three one-dimensional chaotic maps can only keep the chaotic properties within a very small range of control coefficients, while the STL map can stably remain pseudo random in the whole range. The comparison shows that STL maps have better random performance than other maps. In this algorithm, the STL chaotic map is combined with a ZUC stream cipher to enhance its security.

### 2.3. ZUC Stream Cipher

ZUC [39] is a stream cipher named after the Chinese mathematician ZU Chongzhi. It has a series of advantages, such as fast generation of a key stream, a low propagation error rate and easy realization of the hardware circuit. Research proved that the ZUC algorithm has high security and can resist many typical attacks against sequence ciphers. The overall structure of the ZUC algorithm consists of three parts as shown in Figure 3. The input of the ZUC-256 algorithm is a 256-bit initial key(*K*) and 184-bit initial vector(IV). The 32-bit key stream data are generated every cycle.

The specific process is as follows:

(1) Initialization stage: it is necessary to enter the preparation stage before key abortion. The process of sending IV and *K* into the system is called key loading. In this process, *K* and IV are split according to certain rules and spliced together with a set of fixed arrays. Then, start 32 rounds of initialization operations, which can be expressed as:(7)$$\left\{\begin{array}{c} \left(X_{0},X_{1},X_{2},X_{3}\right)=BR\left(S_{15},S_{14},\cdots S_{0}\right)\\ W=F(X_{0},X_{1},X_{2})\\ LFSR\_Initialisation(W>>1) \end{array}\right.,$$
where W>>1 indicates that the high 31 bits of *W* are fed back into the register.

(2) Preparatory work stage: there is no external input in the system at this stage. The difference between the working state and the initialization stage is that LFSR needs to enter the working mode for one iteration; however, the output is abandoned.

(3) Working stage: also called the key stream generation stage. In this stage, the output *W* generated by the nonlinear function and the XOR of the Bit Recombination layer are output as the key stream. The specific operation steps are shown in (Equation 8).
(8)$$\left\{\begin{array}{c} \left(X_{0},X_{1},X_{2},X_{3}\right)=BR\left(S_{15},S_{14},\cdots S_{0}\right)\\ Z=F\left(X_{0},X_{1},X_{2}\right)⊕X_{3}\\ LFSR\_Work(\;) \end{array}\right.,$$

## 3. The Innovative Part of the Encryption Scheme

### 3.1. ZUC Sampling Method

In the process of chaotic image encryption, it is usually necessary to sample the chaotic sequence to reduce the correlation of elements in the chaotic sequence. Equal interval sampling is usually used in the sampling process. The sampling method in this paper is random sampling controlled by the ZUC algorithm, which can further ensure the security. All chaotic sequences in this paper are used after sampling by the ZUC stream cipher system. The specific sampling method is shown in Figure 4.

In the sampling process, the elements in the ZUC sequence and the chaotic sequence are first, corresponded one-to-one, and then the chaotic element corresponding to the 0 element position in the ZUC sequence is discarded.

Logistic, Sine and STL chaotic sequences were sampled, and the Lyapunov exponents of the sequences before and after sampling are shown in Figure 5. The sampling process can improve the Lyapunov exponent of chaotic sequence and improve the randomness of the sequence. The NIST test was also used to test the randomness of the sequence. Prepare 100 groups of original and sampled chaotic sequences, and the length of these sequences is $10^{6}$. The test results are shown in Table 1.

It can be seen that the one-dimensional chaotic sequence used in this paper was sampled, and the LE was improved. NIST tests also show that the randomness of the chaotic system is good. The ZUC sequence used for sampling itself is a stream cipher with strong randomness. The process of sampling improves the problem of the small key space of a one-dimensional chaotic map, making the new chaotic map more suitable for image encryption.

### 3.2. Random Block Method

The image sub-blocks obtained by the ordinary uniform block method still retain the visual information of the original image. The image block process in this algorithm adopts a random block method. When it is necessary to divide an image of size M×N into *K* sub-blocks, there are ${\left(K!\right)}^{M*N/K}$ blocking methods for each image. Take the number of sub-blocks K=4 as an example. The specific steps are as follows:

First, the original image is transformed into a one-dimensional pixel matrix and then divided into groups of every 4 pixels to obtain $\left\{I_{1},I_{2},\cdots ,I_{M\times N/4}\right\}$. A permutation rule library rule is constructed from 24 different permutations of the numbers {1,2,3,4}. The sampled chaotic sequence *S* is mapped to the integers in the interval [1,24] according to the method shown in (Equation 9), corresponding to one line in the rule, respectively. Four pixels in each group were assigned to different image blocks according to the corresponding rules. As shown in (Equation 10).
(9)$$q\left(p\right)=floor(mod\left(S\left(p\right)\times 10^{8},24\right))+1,\;p=1,2,\cdots ,M\times N/4.,$$
(10)$$\left\{\begin{array}{c} {\mathrm{tempB}}_{n}\left(p\right)=I_{p}\left(\mathrm{rule}\left(q\left(p\right),n\right)\right)\\ B_{n}=reshape\left(tempB_{n}\right) \end{array}\right.\;n=1,2,3,4.,$$
where floor is a downward integral function and reshape means to transform the one-dimensional matrix to the desired size. This block method completely separates a group of adjacent pixels. The correlation between adjacent pixels becomes weak, which makes the subsequent encryption better. The Lena image is divided into four and eight sub-blocks as shown in Figure 6.

## 4. The Encryption and Decryption Scheme

As a whole, the algorithm in this chapter divides the image into blocks first and then performs parallel compressed sensing. The chaotic sequences used are all sequences sampled by the ZUC system, and the encryption process is shown in Figure 7.

Step 1: Set the initial vector and initial key of the ZUC stream cipher and run the ZUC system to obtain the binary sequence Z.

Step 2: The discrete wavelet transform (DWT) is used to obtain the image sparse coefficient matrix with a size of M×N. Then, set the initial chaotic value x0 and divide the coefficient matrix into eight blocks $B_{i}$ with a size of m×n using the random block method proposed in Section 3.2. m=M, n=N/8. Eight initial chaotic values are obtained according to (Equation 11).
(11)$$x_{i}=\left(sum\left(B_{i}\right)⊕Z_{i}/10^{8}\right)\;\mathrm{mod}\;1.,$$
where $Z_{i}$ is the ZUC sequence with 8-bit length.i=1,2,⋯8.

Step 3: Set the compression ratio of compression sensing as *r*. According to the chaos initial value calculated above, the chaotic sequence is iterated and sampled by ZUC flow cipher. Eight sequences of length *m* are obtained to construct the measurement matrix of compressed sensing, which is normalized according to (Equation 12):(12)$$C_{i}\left(j\right)=1-{C_{i}}^{2}\left(j\right)\;,\;j=1,2,\cdots mm\times m.$$

The measurement matrix is constituted as follows:(13)$$\Phi_{i}=\sqrt{2/mm}\mathrm{reshape}\left(C_{i},mm,m\right),$$
where mm=round(r×m) is the number of rows of the measurement matrix.

Step 4: Perform compression sensing measurement on block images as shown in (Equation 14) to obtain the compression matrix Y and map it to the interval [0, 255] in accordance with (Equation 15) to facilitate subsequent diffusion operations and obtain the compressed image I.
(14)$$Y_{i}=\Phi_{i}B_{i}.$$

The measurement matrix is constituted as follows:(15)$$I_{i}\left(a,b\right)=\mathrm{round}\left(\frac{255\times (Y_{i}\left(a,b\right)-Y_{\min})}{Y_{\max}-Y_{\min}}\right).$$

Step 5: The compressed image blocks are, respectively, encrypted within the block and encrypted as a whole. The initial values $x_{9}$ and $x_{10}$ are set, and the chaotic sequences are iterated, respectively. Both sequences are sampled by ZUC to obtain $C_{9}$ and $C_{10}$. $C_{9}$ is used to diffuse and scramble the block image. Then, the image blocks are reassembled, and $C_{10}$ is used to carry out the whole diffusion and scrambling operation again in order to avoid the block effect.

The decryption process of this algorithm is shown in Figure 8.

(1) The decryption process of this algorithm is the reverse process of encryption. The receiver first performs inverse diffusion and scrambling on the encrypted image according to the key to obtain the compressed image.

(2) Inverse mapping of the compressed image according to (Equation 16). Then, eight measurement matrices are constructed according to the key, and each image block is reconstructed. In this paper, the OMP algorithm is chosen as the reconstruction algorithm.
(16)$$Y_{i}\left(a,b\right)=\frac{I_{i}\left(a,b\right)\times \left(Y_{\max}-Y_{\min}\right)}{255}+Y_{\min},$$

(3) Finally, the sparse coefficient matrix of the image is obtained by re-partitioning the image. The original image can be obtained by inverse DWT.

This algorithm is suitable for both grayscale and color images. When encrypting a color image, it only needs to be decomposed into the three components of RGB. When preparing the chaotic sequence required for encryption, the sequence length can be three times the original.

## 5. Simulation Results and Performance Analysis

This section will simulate the performance of the proposed algorithm and analyze the results. Experiments are conducted on MATLAB R2014a in a computer with Intel Core i5 2.6 GHz and 8 GB RAM. In this algorithm, compressive sensing is used to realize image compression, and the measurement matrix consists of STL chaotic sequences sampled by the ZUC system. In the decryption process, the OMP algorithm is used to reconstruct the image. The initial key and initial value vector of the ZUC system are all zero. Set the control parameter of chaotic system μ=4.1, initial value x0=0.5489 and compression ratio r=0.5.

Figure 9 shows the encryption and decryption results of color images and grayscale images. It can be seen from the image in the second line that the size of the image processed by the algorithm is compressed. The encrypted image is similar to the noise image, and the information of the original image cannot be observed subjectively. It can be seen from the decrypted image in the third row that the algorithm achieves good reconstruction of the image.

### 5.1. Histogram Analysis

As the original image contains certain information, the values of some pixels will be close to or even the same. The gray histogram will be uneven and will contain certain statistical information. For a good encryption algorithm to resist statistical analysis, the image pixels should be evenly spread through the gray range [0, 255].

Figure 10 shows the histograms of the original, encrypted and decrypted images for the R, G and B components of the color Lena image. Figure 11 shows the histograms of the two grayscale images before and after encryption. It can be seen that the number of pixels at each gray value in the encrypted image is basically the same in the interval [0,255], and thus the stealer cannot obtain the original image through the statistical information of the encrypted image.

As compressed sensing is a lossy compression technology; some details contained in the original image are lost in the reconstruction process, and the reconstructed image is not completely consistent with the original image histogram. It can be seen from the above histogram that several peak values of the histogram reconstructed for Lena are reduced, and the histogram becomes flat by comparing the first and third rows in the diagram. However, it can be seen in Figure 9 that the reconstruction of the decrypted image is still successfully achieved intuitively according to the analysis.

### 5.2. Pixel Correlation Analysis

The correlation distribution of the original adjacent pixels is shown in Figure 12a. It can be seen that the values of adjacent pixels are very similar with strong correlations no matter in which direction. The evenly distributed pixels of the encrypted image in Figure 12b indicate that the pixels of the ciphertext image have been scrambled. The encryption requirements are met from the perspective of pixel correlation.

We calculated the correlation coefficient of adjacent pixels of the encrypted image and compared it with other algorithms that use compressed sensing technology to achieve image encryption. The results are shown in Table 2. The correlation of adjacent pixels in the image processed by the algorithm in this paper is lower than that of the other two algorithms in the horizontal and vertical directions, and the pixel correlation is generally lower than the other two algorithms. The pixel confusion effect of this algorithm is better, the security is higher, and the ability to resist malicious analysis is stronger.

### 5.3. Information Entropy Analysis

Information entropy can calculate the uniformity of gray distribution by measuring the probability of gray appearance. The calculation formula is:(17)$$H\left(m\right)=-\sum_{i=1}^{n}p\left(m_{i}\right)\log_{2}p\left(m_{i}\right),$$
where *m* is the grayscale set of image pixels. $p(m_{i})$ is the probability of $m_{i}$ occurring. *n* is the total number of $m_{i}$.

In order to explore the information entropy of the image encrypted by this algorithm and the influence of the compression rate on the information entropy value, Table 3 shows the information entropy of three different encrypted images and calculates them by changing the compression rates. It can be seen that the encrypted images with different compression rates maintain high entropy, and the uniformity of the gray distribution is not affected by the compression operation, indicating that the algorithm can resist information entropy attacks.

### 5.4. Key Sensitivity Analysis

The key sensitivity in this section is analyzed from two aspects. On the one hand, the encryption key is slightly changed, and the changes of the encrypted image before and after the change are observed. On the other hand, in the decryption process, we make a slight adjustment on the basis of the correct key, use the wrong key to decrypt and observe whether the decryption can be correct. During the encryption process, the encryption key x0 is slightly changed (order of magnitude $10^{-16}$). The encrypted images before and after the key change and the difference images of the two images are shown in Figure 13.

In order to quantify the difference degree of the encrypted image before and after the key change, the mean structural similarity (MSSIM) of the two images is calculated. The smaller the MSSIM value is, the greater the influence of key changes on the encryption result is, and the higher the encryption key sensitivity of the algorithm is. The calculation formula is shown in (Equation 18).
(18)$$MSSIM=\frac{1}{N}\sum_{i=1}^{N}\frac{\left(2\mu_{x}\mu_{y}+c_{1}\right)\left(2\sigma_{xy}+c_{2}\right)}{\left({\mu_{2}}^{2}+{\mu_{y}}^{2}+c_{1}\right)\left({\sigma_{x}}^{2}+{\sigma_{y}}^{2}+c_{2}\right)},$$
where $\mu_{x}$, $\mu_{y}$, ${\sigma_{x}}^{2}$ and ${\sigma_{x}}^{2}$ are the mean and variance of *x* and *y*, respectively. $\sigma_{xy}$ is the covariance of Windows *x* and *y*. *N* is the total number of Windows for the image. $c_{1}={\left(0.01\times 255\right)}^{2}$ and $c_{1}={\left(0.03\times 255\right)}^{2}$. The MSSIM of Figure 13b,c was calculated as $-5.9023\times 10^{-4}$. The Hamming distance is used in data transmission error control coding, which indicates the number of different characters in the corresponding positions of two strings of the same length. We converted the two images into binary vectors, and the calculated Hamming distance was 0.4999. Different encryption results were obtained when the key was infinitesimally changed. The algorithm has good encryption key sensitivity.

The decryption key sensitivity of the algorithm is analyzed below. In the decryption stage, a small disturbance is added on the basis of the correct decryption key to obtain a new encrypted image. Figure 14 shows the wrong decrypted images after making some slight changes in the decryption key. It can be seen that, even if there is such a small deviation of the key used in encryption and decryption, the decryption result is still completely wrong. The proposed algorithm is highly sensitive regarding the decryption key.

### 5.5. Key Space Analysis

The contents of the key in this algorithm include 10 chaotic initial values, the initial vector and the initial key of the ZUC system. The calculation accuracy is $2^{-52}$ according to the IEEE745 standard. The key space is $2^{52\times 10}\times 2^{184+256}=2^{960}$. This algorithm expands the key space by introducing the ZUC stream cipher.

Table 4 shows that the key space of this algorithm is much larger than that of the other five algorithms. The stronger the resistance of this scheme to exhaustive analysis, the higher the security.

### 5.6. Differential Attack Analysis

Avalanche effect analysis studies the impact of changes in the input information of encrypted images. There are two important indicators: the pixel number change rate (NPCR) and the unified average intensity change (UACI). Inputting different plaintext images and calculating the NPCR can quantify the number of pixels that have been changed, and UACI is biased towards calculating the intensity of the changes in pixel values at corresponding positions. The calculation formulas are as follows:(19)$$NPCR=\frac{1}{N\times M}\sum_{i=1}^{N}\sum_{i=1}^{M}E(i,j),$$
(20)$$UACI=\frac{1}{N\times M}\sum_{i=1}^{N}\sum_{j=1}^{M}\frac{\left|M_{1}\left(i,j\right)-M_{2}\left(i,j\right)\right|}{255},$$
where $M_{1}$ is a normal encrypted image, and $M_{2}$ is an encrypted image after changing a pixel in the original image. E(i,j)=0 when the pixel values of the two images are the same, otherwise E(i,j)=1. A pixel value in the original image was randomly changed, and the values of NPCR and UACI before and after the change were calculated as shown in Table 5.

Compared with [40,41,42,44], the values of NPCR and UACI are closer to the expected values. Our algorithm is more sensitive to small changes in the original image. A random pixel change in the plaintext image makes the encrypted image a completely different image when compared with the original one. The algorithm has good performance in resisting differential attacks.

### 5.7. Analysis of the Reconstruction Effect

The compressed sensing operation loses image information to a certain extent. The reconstruction effect of quantified images can be calculated by calculating the peak signal-to-noise ratio (PSNR). The mean square error (MSE) of the original image and the reconstructed image is regarded as a noise signal, and the PSNR is the ratio of the maximum possible power to the mean square error of the two images. The formula is as follows:(21)$$PSNR=10\log_{10}\left(\frac{{\left(2^{8}-1\right)}^{2}}{MSE}\right).$$

In order to verify the effect of the ZUC stream cipher on the image reconstruction effect of the chaotic sequence sampling operation, the decrypted image obtained is directly reconstructed without the mapping step of the compressed matrix. The original chaotic sequence and the sampled chaotic sequence are used to construct the measurement matrix of compressed sensing, and the PSNR and MSSIM of the reconstructed image and the original image are compared.

In Table 6, using the chaotic sequence sampled by the ZUC stream cipher to construct the measurement matrix can not only make the chaotic sequence more difficult to predict but also improve the image reconstruction effect to a certain extent.

The following image encryption is performed according to the algorithm proposed in this paper. We used multiple images with a size of 512×512 for encryption and decryption to calculate the PSNR of the original image and the decrypted image. It can be seen in Table 7 that the PSNR of the image is above 27dB, and the MMSIM is above 0.84.

Then, the original image is replaced with an image of size 256×256, which is compressed and reconstructed. Table 8 shows the PSNR of different decrypted images, which are compared with [40,41,44]. It can be seen by comparison that the PSNR value of the algorithm in this paper is relatively high. The reconstruction effect of this algorithm is good, and the distortion caused by encryption and decryption is small.

In order to compare the effect of the compression ratio on the image reconstruction effect, the PSNR under different compression ratios was obtained and compared with [43]. Figure 15 shows that the proposed algorithm has a better image reconstruction effect when the compression ratio is less than 0.8. It shows that the algorithm can compress the image into a smaller image and achieve good reconstruction.

### 5.8. Analysis of Encryption and Decryption Efficiency

In the evaluation of encryption algorithms, security and efficiency are two important indicators. The security of this algorithm has been explained in several aspects. For images of different sizes, the encryption times required by this algorithm are shown Figure 16. The encryption time required by this algorithm is largely less than that [40], which is very close to the time required by [43]. When the image size is less than 256×256, the encryption time is less than 0.6 s. The comparison results of the encryption time and decryption time are shown in Table 9. Compared with [41], this algorithm spends less time in the decryption phase.

It can be seen from the comparison that, since the algorithm adopts the block-parallel compressed sensing method to process the image, the encryption and decryption time is less. Even if the ZUC algorithm is introduced, some elements in the chaotic sequence need to be lost in the sampling process; however, there is no obvious loss of efficiency.

## 6. Conclusions

In this study, we aimed to encrypt images while saving communication channel resources, and we proposed an image block compression and encryption algorithm based on the theory of chaotic compressed sensing. A ZUC stream cipher was used to improve the deficiencies of a one-dimensional chaotic system. This algorithm randomly divides the sparse coefficient matrix of the image into blocks and uses the chaotic sequence sampled by the ZUC stream cipher to construct the measurement matrix.

Simulation experiments and comparison with other algorithms showed that the key space of the proposed algorithm was large and that ZUC sampling improved the image reconstruction effect. This paper also analyzed the security under different compression ratios from different aspects and compared it with other algorithms, which proved that the algorithm had high security.

## Figures and Tables

**Figure 1 entropy-24-00742-f001:**
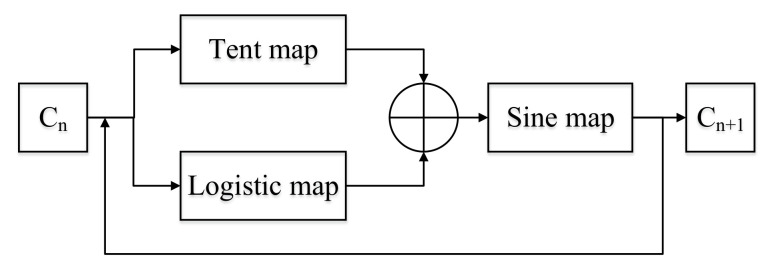
STL map structure.

**Figure 2 entropy-24-00742-f002:**
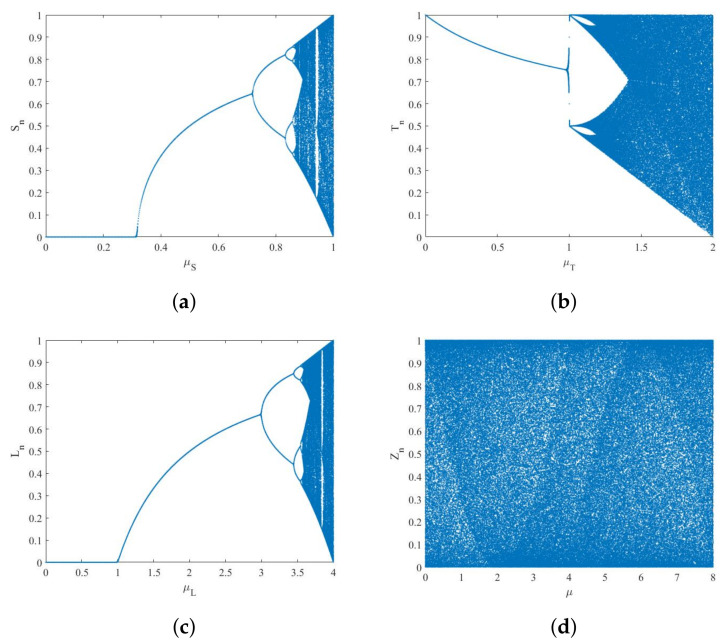
Bifurcation diagram: (**a**): Sine map. (**b**): Tent map. (**c**): Logistic map. (**d**): STL map.

**Figure 3 entropy-24-00742-f003:**
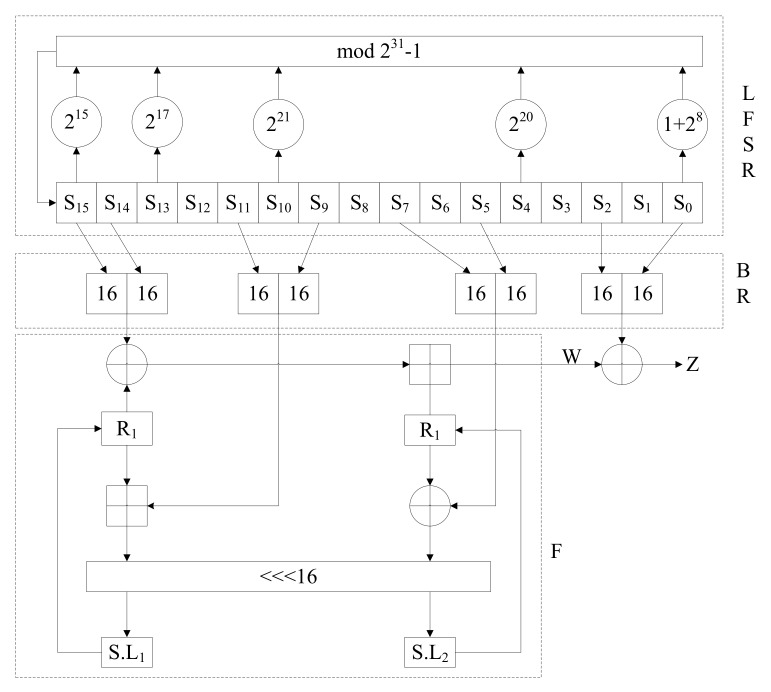
ZUC system structure.

**Figure 4 entropy-24-00742-f004:**
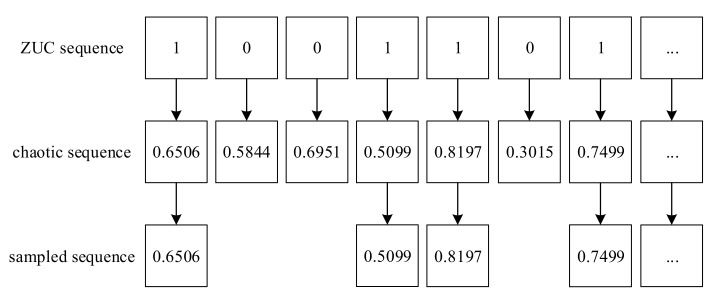
The sampling process diagram.

**Figure 5 entropy-24-00742-f005:**
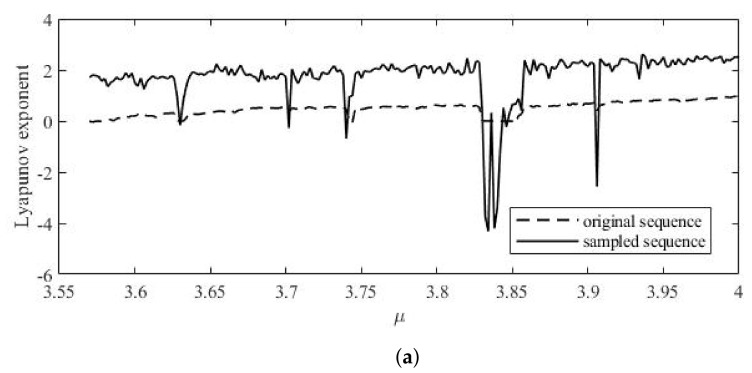
Lyapunov exponents: (**a**): Logistic map. (**b**): Sine map. (**c**): STL map.

**Figure 6 entropy-24-00742-f006:**
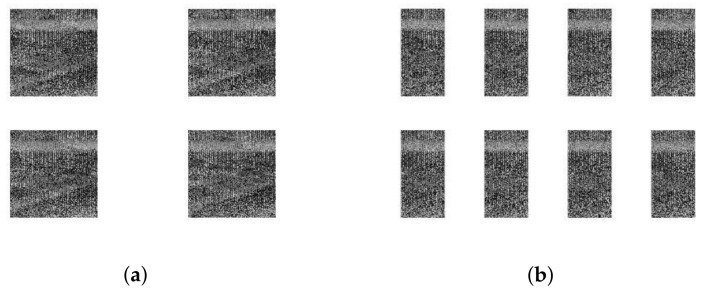
Random block rendering: (**a**): 4 sub-blocks. (**b**): 8 sub-blocks.

**Figure 7 entropy-24-00742-f007:**
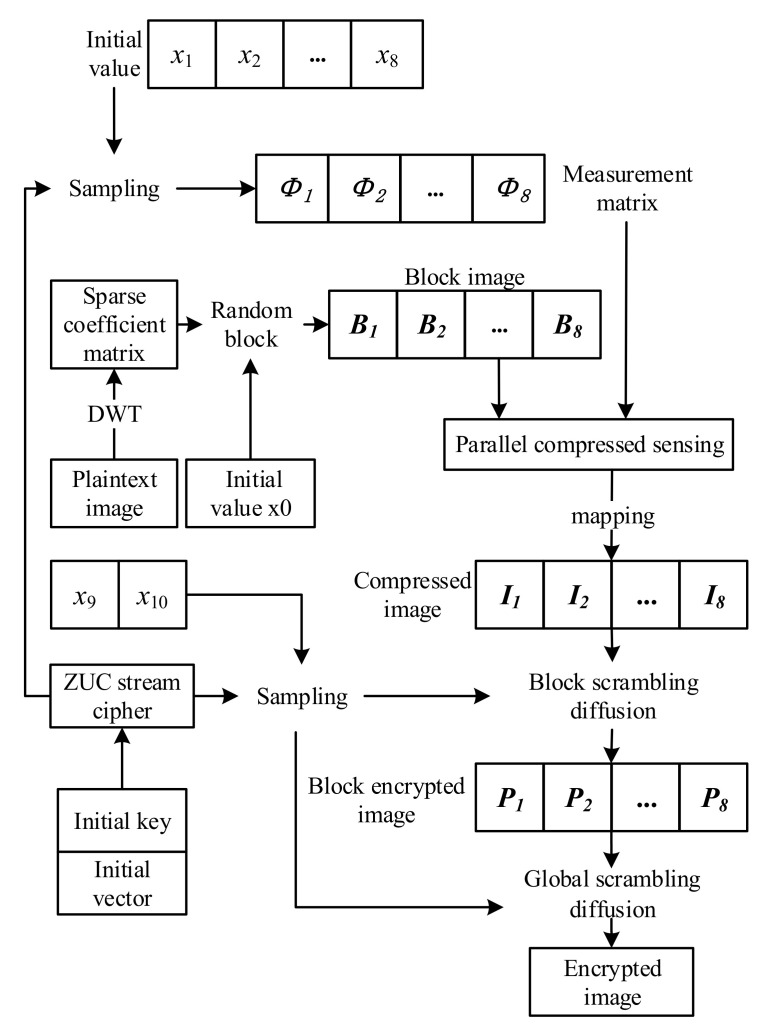
Encryption flow chart.

**Figure 8 entropy-24-00742-f008:**
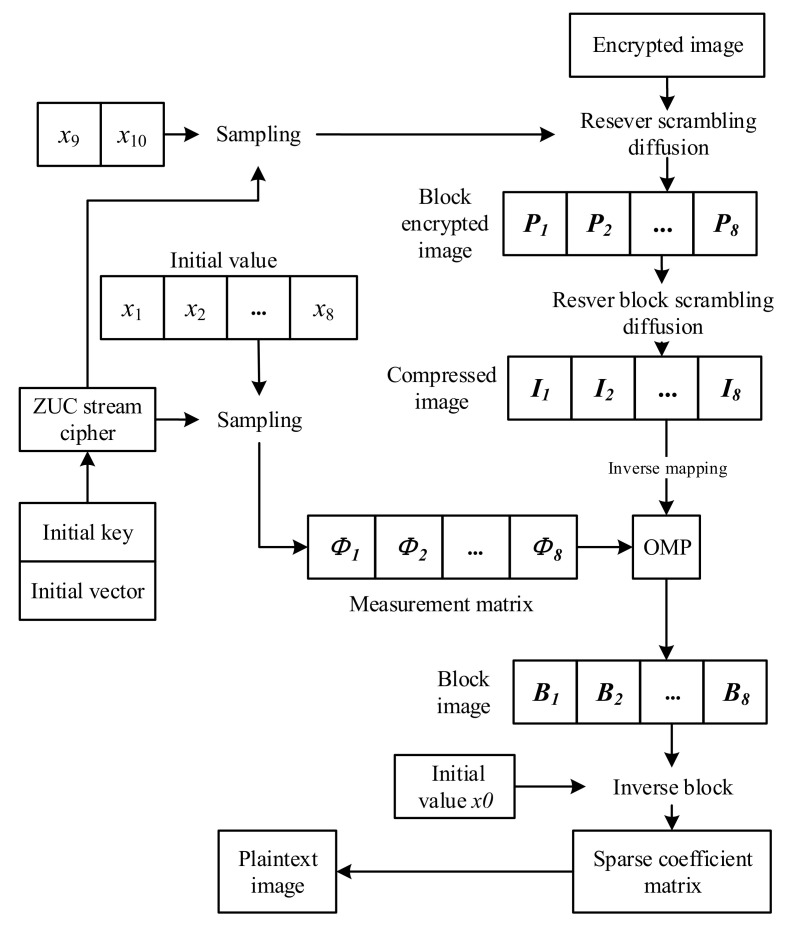
Decryption flow chart.

**Figure 9 entropy-24-00742-f009:**
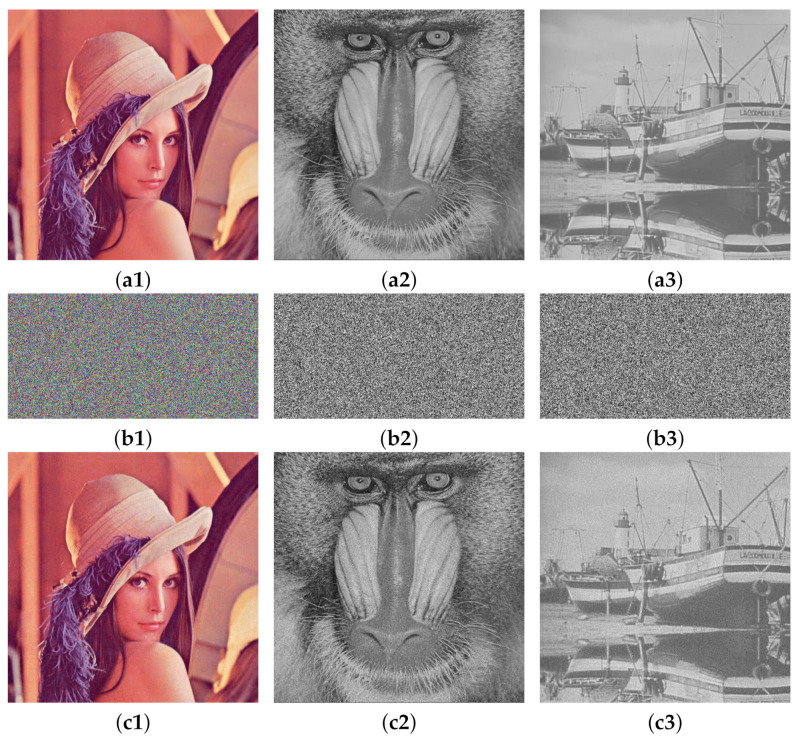
Simulation results of encryption and decryption: (**a1**)–(**a3**) are plain images Lena, Baboon and Boat; (**b1**)–(**b3**) are the encryption images; and (**c1**)–(**c3**) are the decrypted images.

**Figure 10 entropy-24-00742-f010:**
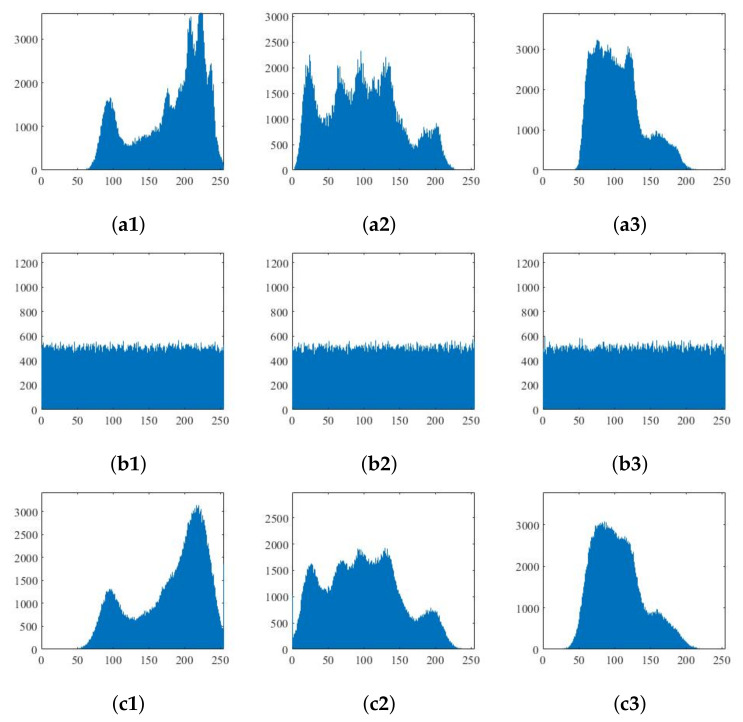
Histogram of the RGB components of Lena image: (**a1**)–(**a3**) are the plaintext images; (**b1**)–(**b3**) are the encryption images; and (**c1**)–(**c3**) are the decrypted images.

**Figure 11 entropy-24-00742-f011:**
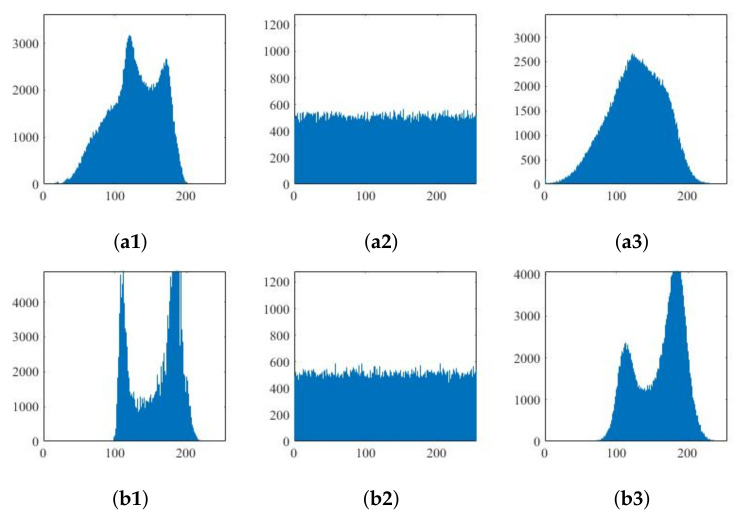
Histogram of grayscale image before and after encryption: (**a1**)–(**a3**) Baboon and (**b1**)–(**b3**) Boat.

**Figure 12 entropy-24-00742-f012:**
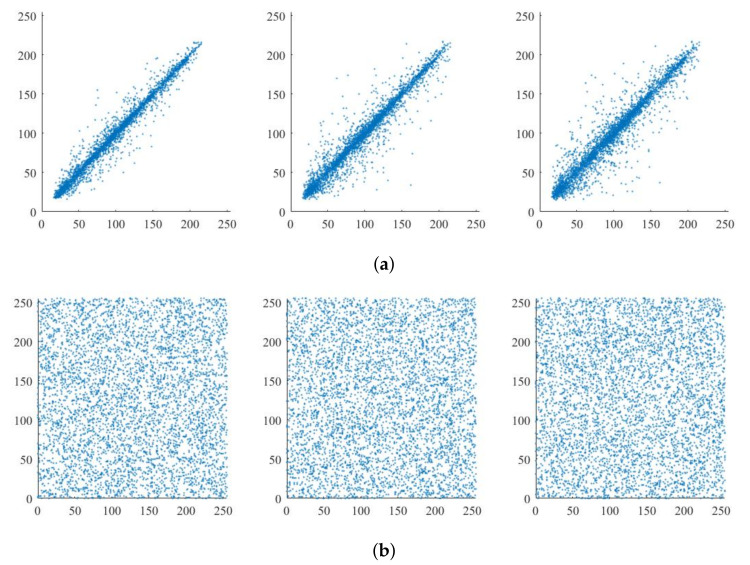
Pixel correlation distribution: (**a**): Original image. (**b**): Encryption image.

**Figure 13 entropy-24-00742-f013:**
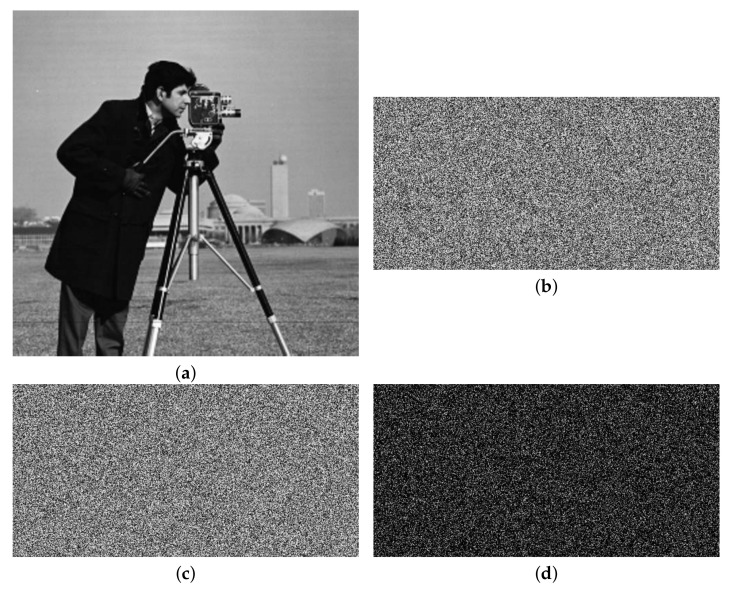
Encryption key sensitivity: (**a**): The original image. (**b**): Encryption image. (**c**): The encrypted image after changing the key. (**d**): The difference between two encrypted images.

**Figure 14 entropy-24-00742-f014:**
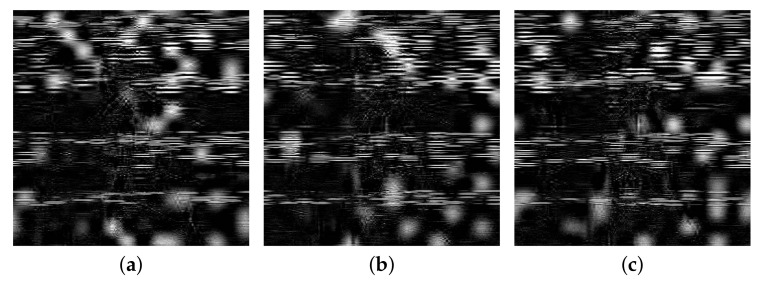
Incorrectly decrypted image: (**a**): key changed $10^{-14}$, (**b**): key changed $10^{-15}$ and (**c**): key changed $10^{-16}$.

**Figure 15 entropy-24-00742-f015:**
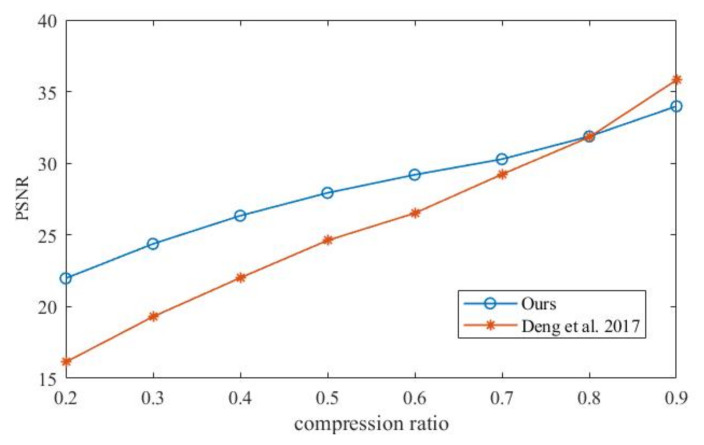
PSNR comparison (ours vs. Deng et al. 2017 [43]) under different compression ratios.

**Figure 16 entropy-24-00742-f016:**
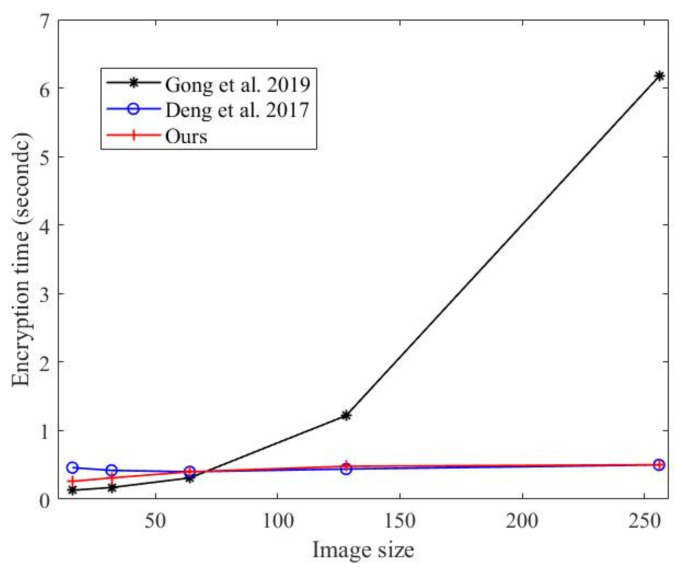
Encryption time comparison (ours vs. Gong et al. 2019 [40], Deng et al. 2017 [43]).

**Table 1 entropy-24-00742-t001:** NIST test results of the sequences before and after being sampled.

Test	Original Sequence		Sampled Sequence	
	*p*-Value	Passing Rate	*p*-Value	Passing Rate
Frequency test	0.4682	100	0.4767	100
Block-frequency test	0.4727	99	0.4579	100
Runs test	0.4497	98	0.4262	98
Longest Run test	0.5572	97	0.5357	98
Rank test	0.1129	95	0.1865	98
FFT test	0.5237	99	0.4829	99
Non-Overlapping Templates	0.6182	100	0.5240	99
Overlapping-templates	0.5011	98	0.4291	98
Universial	0.5020	96	0.4542	99
Linear Complexity	0.5499	100	0.4517	100
Serial Test	0.2773	98	0.2722	97
Approximate Entropy	0.5121	99	0.5083	100
Cumulative Sums Test	0.7721	98	0.7655	100
Random Excursions	0.5207	99.32	0.4873	99.57
Random Excursions Variant	0.4883	99.12	0.4998	99.57

**Table 2 entropy-24-00742-t002:** Correlation coefficient of adjacent pixels.

Image	Method	Horizontal Direction	Vertical Direction	Diagonal Direction
Peppers	Ours	−0.0037	0.0021	−0.0018
	[40]	−0.0068	0.0091	−0.0061
	[41]	0.0234	0.0121	0.0005
	[42]	0.0037	−0.0014	0.0073
Cameraman	Ours	0.0032	0.0042	−0.0068
	[40]	−0.0035	0.0049	−0.0099
	[41]	0.0062	−0.0054	−0.0014
	[42]	0.0064	−0.0082	0.0091
Woman	Ours	0.0032	0.0052	−0.0068
	[40]	0.0016	0.0081	−0.0016
	[43]	0.0950	0.2432	0.0152

**Table 3 entropy-24-00742-t003:** Information entropy of encrypted images.

Image	Compression Ratio			
	0.3	0.5	0.7	0.9
Lena	7.9960	7.9961	7.9965	7.9972
Peppers	7.9952	7.9953	7.9965	7.9961
Woman	7.9941	7.9960	7.9969	7.9976

**Table 4 entropy-24-00742-t004:** Key space comparison.

	Ours	[40]	[41]	[43]	[42]	[44]
Key space	$2^{960}$	$2^{176}$	$2^{399}$	$10^{37}$	$2^{626}$	$2^{200}$

**Table 5 entropy-24-00742-t005:** NPCR and UACI.

	Expected Value	Ours	[40]	[41]	[42]	[44]
NPCR(%)	99.6094	99.6043	99.6155	99.4700	99.65	99.59
UACI(%)	33.4635	33.4603	33.5595	33.1800	33.64	33.51

**Table 6 entropy-24-00742-t006:** Comparison of reconstruction effects before and after sampling.

Plain Image	PSNR (dB)		MSSIM	
	Original	Sampled	Original	Sampled
Lena	34.6021	34.9111	0.9279	0.9285
Pepper	37.8777	38.3088	0.9527	0.9561
Cameraman	31.3254	31.4581	0.8496	0.8518
Woman	42.5347	42.6238	0.9768	0.9775

**Table 7 entropy-24-00742-t007:** Reconstruction effect of 512 × 512 images.

	Peppers	Woman	Cameraman	Lake	Boat	Lena
PSNR(dB)	30.7601	31.5932	28.6057	27.5064	28.3154	31.3899
MSSIM	0.9077	0.9220	0.8436	0.8536	0.8591	0.9209

**Table 8 entropy-24-00742-t008:** PSNR comparison table for 256×256 images.

Image	Ours	[40]	[41]	[44]
Peppers	29.6180	24.8485	-	23.0676
Woman	30.9270	30.8184	23.5210	-
Cameraman	26.2482	-	25.2684	23.8783

**Table 9 entropy-24-00742-t009:** Encryption and decryption time of the proposed scheme (s).

Image Size	Encryption Time		Decryption Time	
	Ours	[41]	Ours	[41]
256×256	0.5028	0.9039	3.5124	9.992
512×512	0.7727	1.2688	6.4579	35.280
1024×1024	1.4497	2.5490	15.4262	159.500

## Data Availability

Not applicable.

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
