# Peer review of "An Image Compression Encryption Algorithm Based on Chaos and ZUC Stream Cipher"

_entropy, 2022, doi:10.3390/e24050742_

Round 1

Reviewer 1 Report

The authors proposed image encryption based on a chaotic map. The main encryption structure is focused on grayscale images, which makes the applicability of the proposal work limited. I invite the authors to revise their proposal and extend the work to cover RGB images. It is also required to have a time execution test or complexity evaluation.

The paper contains poor comparative studies. The authors should evaluate and compare their results further with recent image encryption schemes.

Please proofread the manuscript.

Author Response

Thanks for your comment. Please see the attachment.

Reviewer 2 Report

General Comments:

Any cryptographic system used in cryptography applications must meet two requirements: security and computational complexity.

The statistical security of the proposed system is studied but its security against cryptographic attacks, especially physical attacks (SCA, FIA), is not studied. Indeed, ZUC algorithm presents some weaknesses against SCA, see [1].

The computational performance (ET, NCpB) of the system should be carried out and compared to some systems in the literature. To compare the running speed of different systems working on different platforms usually we use the NCpB (Number of needed Cycles to generate one Byte) parameter witch is defined as:

ET(Mbit/s)=(Image size (Mbit))/(Average generation time (s))

NCpB=(CPU Speed (Hz))/(ET (Byte⁄s))

In order to simplify the complexity of your system, you can replace the ZUC algorithm by simple chaotic mixing based on the used one-dimensional Sine-Tent-Logistic (STL) chaotic system, see [2].

Comments:

  • There are some ambiguity on the used symbols, e.g.:
  • In paragraph 2.1, you wrote: “The image of size M X M is divided into image blocks xi of size M X N. But in paragraph 3.2, you wrote “to divide an image of size M X N into K sub-blocks.
  • In order to avoid ambiguity between the output Z of ZUC algorithm and the output Z of STL map, replace in figure 1, Zn by Cn
  • 5 lines below figure 4, Replace: “The sampling method in this paper is random sampling controlled by the chaotic system”

 By: “The sampling method in this paper is random sampling controlled by the

ZUC algorithm”

  • In relation (9), the sampled sequence S(p) is used and not C(p)
  • It is better to apply NIST test instead of Lyapunov exponents on the output sequences of Logistic, Sine map and STL map (of figure 5).
  • Figure 6, is not clear and must be replaced
  • The better parameter to quantify the Key sensitivity is the Hamming Distance.

[1] TANG Ming et al., 2012,. "Differential Power Analysis on ZUC Algorithm."

[2] O. Jallouli, et al., 2017, “Design and Analysis of two Stream Ciphers Based on Chaotic Coupling and Multiplexing techniques” MTAP, Multimedia Tools and Applications, June 2017, pp. 1-27. DOI 10.1007/s11042-017-4953-x

Author Response

Thank you for your valuable. Please see attachment.

Reviewer 3 Report

In this paper, authors proposed a novel image encryption algorithm based on ZUC stream cipher and compressed sensing. The sampled chaotic sequence and the parallel compressed sensing operation have some new ideas. It is of reference significance for cryptosystem designers.  I have the following  comments for reference.
1. In the Abstract, the first sentence is too long and has word repetition with use of "based on" and "on
the basis of".
2. In Eq.(5), "S∈[0,1]"should be written as "S_n∈[0,1], n=0,1,...", T and L also have such problems, .
3. For Eq.(6), the source references should be cited.
4. In Figure 4, the text size should be the same. "the digits at the position Z of ZUC sequence 1 corresponding
to the original chaotic sequence C are retained", the expression of this sentence is not good. 
5. The title of Figure 8 should be "Decryption flow chart".
6. "Figure 9 shows the result of ..." should be "Figure 9 shows the results of ...". "It can be seen from Figure 10 that In the interval [0,255],..." should be  "It can be seen from Figure 10 that, in the interval [0,255], ... ". It is suggested that the author carefully check the language of the full text.
7. In Figure 9, the title of "(b1)-(b3) are the encryption image images" has two words of "image".
8. The abscissa and ordinate axes in Figure 14 shall be marked with coordinate quantity symbols.
In my opinion, the manuscript may be accepted for publication after minor revision.

Author Response

(The authors gave the same response as above.)

Round 2

Reviewer 1 Report

I have no further comments

Reviewer 2 Report

Your responses to my general comments are not satisfied